# Model Selection for Production System via Automated Online Experiments

**Zhenwen Dai**
Spotify
zhenwend@spotify.com

**Praveen Ravichandran**
Spotify
praveenr@spotify.com

**Ghazal Fazelnia**
Spotify
ghazalf@spotify.com

**Ben Carterette**
Spotify
benjaminc@spotify.com

**Mounia Lalmas-Roelleke**
Spotify
mounial@spotify.com

## Abstract

A challenge that machine learning practitioners in the industry face is the task of selecting the best model to deploy in production. As a model is often an intermediate component of a production system, online controlled experiments such as A/B tests yield the most reliable estimation of the effectiveness of the whole system, but can only compare two or a few models due to budget constraints. We propose an automated online experimentation mechanism that can efficiently perform model selection from a large pool of models with a small number of online experiments. We derive the probability distribution of the metric of interest that contains the model uncertainty from our Bayesian surrogate model trained using historical logs. Our method efficiently identifies the best model by sequentially selecting and deploying a list of models from the candidate set that balance exploration-exploitation. Using simulations based on real data, we demonstrate the effectiveness of our method on two different tasks.

## 1 Introduction

Evaluating the effect of individual changes to machine learning (ML) systems such as choice of algorithms, features, *etc.*, is the key to growth in many internet services and industrial applications. Practitioners are faced with the decision of choosing one model from several candidates to deploy in production. This can be viewed as a model selection problem. Classical model selection paradigms such as cross-validation consider ML models in isolation and focus on selecting the model with the best predictive power on unseen data. This approach does not work well for modern industrial ML systems, as such a system usually consists of many individual components and a ML model is only one of them. The metric of interest often depends on uncontrollable factors such as users' responses. Only optimizing the predictive power of the ML model would not lead to a better metric of the overall system. Instead, randomized experiments (also known as "A/B tests") are considered as the gold-standard for evaluating system changes [1] as they provide a more direct measure of the metric. However, only a few variants of the ML model can be tested using randomized experiments as they are time-consuming to conduct and have resource constraints (such as the number of active users, *etc.*). Furthermore, deploying bad systems can lead to catastrophic consequences.

An alternative approach is to exploit log data collected under the production system to estimate the metric of interest if a different ML model is deployed. Typical methods include developing offline measures or simulators that model users' behavior [2, 3], and replaying the recorded decisions with a probability ratio correction between the new and previous model, which is referred to as off-policy evaluation (OPE) [4, 5, 6, 7, 8, 9]. A big challenge faced by these methods is the selection bias of the

log data. As a consequence, these methods work well when the considered model behaves similar to the logging model, but the effectiveness deteriorates quickly when the considered model behaves increasingly differently from the logging model.

To overcome the selection bias, we suggest to include the data collection process into the model selection approach. We propose a new framework of model selection for production system, where the best model is selected via deploying a sequence of models online. This allows deploying the model that can provide maximum information, iteratively refining the belief about the candidate models and efficiently identifying the model that leads to the best metric for the overall system. Concretely, we target at a specific but widely existing scenario, in which the metric of interest can be decomposed into an average of immediate feedback, *e.g.*, the click-through rate in recommender systems. We develop a Bayesian surrogate model that can efficiently digest the collected data from online experiments and derive the probability distribution of the metric of interest based on the surrogate model. The model to deploy is selected by balancing the exploration-exploitation trade-off. Comparing with A/B testing, our approach can perform model selection from a large pool of candidates by using not only the recorded metric but also the log data about individual user interactions. Comparing with OPE, our approach provides more accurate estimation of model performance by avoiding the selection bias through controlling the data collection process. Overall, our approach correctly identifies the best model even if it behaves very differently from the one in production.

## 2 Model selection with automated online experiments

We define the problem of model selection for production system (MSPS) as follows: given a set of candidate models $\mathbf{M}_i \in \mathcal{M}$ and an online budget, the goal is to identify model $\mathbf{M}^*$ with maximum utility of the overall system:

$$\mathbf{M}^* = \arg \max_{\mathbf{M}_i \in \mathcal{M}} v(\mathbf{M}_i). \tag{1}$$

In this work, we focus on the scenario where a model takes an input representation $\mathbf{x}$, returns a decision $\mathbf{a}$ while observing an immediate feedback for each individual decision. The utility associated with a given model $\mathbf{M}_i$ is influenced by immediate feedback, which could be an indirect and complex relationship such as the relation between profit margin and user clicks. Here, we restrict our focus to the cases where the utility has an additive relation with immediate feedback and refer to it as *accumulative metric*. The above setting is common in the industry; for example, in recommender systems, the inputs $\mathbf{x}$ are users or context (user representation, time of the request, *etc.*), the decisions are the choice of recommendation, and the accumulative metric could be a metric such as total consumption, which is the sum of consumption associated with individual recommendations.

A model can be represented as a probability distribution of the decision conditioned on the input $p(\mathbf{a}|\mathbf{x})$, where a deterministic system simply results in a delta distribution. The distribution of the inputs to the model represented as $p(\mathbf{x})$ is typically unknown. The accumulative metric for a given model $\mathbf{M}_i$ can be defined as,

$$v(\mathbf{M}_i) = \int_{\mathcal{X}} \int_{\mathcal{A}} m \, p(m|\mathbf{a}, \mathbf{x}) p(\mathbf{a}|\mathbf{x}, \mathbf{M}_i) p(\mathbf{x}) \, \mathrm{d}\mathbf{a} \, \mathrm{d}\mathbf{x}, \tag{2}$$

where the integration is over the space of input $\mathbf{x} \in \mathcal{X}$ and the space of decision $\mathbf{a} \in \mathcal{A}$. The accumulative metric is defined as an expectation of immediate feedback with respect to the distribution of input and decisions conditioned on individual inputs. Unfortunately, the accumulative metric is not tractable, because both the distribution of input $p(\mathbf{x})$ and the distribution of immediate feedback $p(m|\mathbf{a}, \mathbf{x})$ are unknown.

With a production system, the information about the accumulative metric can be collected by deploying the model of interest in production and let it react to real traffic and record the corresponding accumulative metric. The collected data from such a deploy consist of a recorded accumulative metric, in our case $\hat{v} = \frac{1}{N} \sum_i m_i$, and a set of interactions $\mathcal{D} = \{(m_i, \mathbf{a}_i, \mathbf{x}_i)\}_{i=1}^N$. In this work, we define MSPS as a sequential optimal decision problem. A MSPS method iteratively chooses a model from the candidate set to deploy online for data collection with the aim of identifying the model with the best accumulative metric in the fewest number of online experiments. A model deployment is a expensive process, as each deployment takes a long time, and only a small number of models can be deployed in parallel due to the limited number of users and the affordable degradation in service quality. Global optimization methods like Bayesian optimization (BO) [10] do not work well in this

setting, because BO requires the search space to be represented in a relatively low dimensional space but embedding the model candidate sets (especially models of different types) into a low-dimensional space is non-trivial. Unlike BO methods that only take into account the accumulative metric from online experiments, our approach takes advantage of the full log data by training a Bayesian surrogate model. The uncertainty of the surrogate model is then used to balance between exploring an uncertain but potentially good choice and exploiting a known one.

## 3  Bayesian surrogate for accumulative metric

Instead of using the recorded accumulative metric from online experiments, we propose to estimate it from its definition in (2). In this formulation, $p(\mathbf{a}|\mathbf{x}, \mathbf{M}_i)$ is known and $p(\mathbf{x})$ could be replaced with an empirical distribution, therefore, the key is to capture the distribution of the immediate feedback $p(m|\mathbf{a}, \mathbf{x})$. The data collected from online experiments contains lots of data points about this distribution. This allows us to build a Bayesian surrogate model for the immediate feedback.

### 3.1  Gaussian process surrogate model

We propose to use a Gaussian process (GP) as the surrogate model for the distribution of the immediate feedback. There is often stochasticity in the immediate feedback data including the intrinsic stochasticity in human interactions, e.g., some random reactions from a user, as well as the information that is not available to the production system. To accommodate this stochasticity, we divide the Bayesian surrogate model into two parts: (i) a GP model that captures the "noise-free" component of the immediate feedback, denoted as $p(f|\mathbf{a}, \mathbf{x})$; (ii) a noise distribution used to absorb all the stochasticity that cannot be explained by $\mathbf{x}$ and $\mathbf{a}$, denoted as $p(m|f)$. When the immediate feedback is a continuous value, we use a Gaussian noise distribution. The resulting surrogate model can be written as,

$$m = f(\mathbf{a}, \mathbf{x}) + \epsilon, \quad f \sim \mathcal{GP}(0, k(\cdot, \cdot)), \quad \epsilon \sim \mathcal{N}(0, \sigma^2), \tag{3}$$

where the GP has zero mean and a covariance function $k(\cdot, \cdot)$. Stationary covariance functions are the most common choices, such as the radial basis function (RBF) and the Matérn covariance functions. Note that the distribution of the immediate feedback $p(m|\mathbf{a}, \mathbf{x})$ is independent of the choice of candidate models. This allows us to train a single surrogate model and use it to score all the candidate models.

In some use cases, the inputs $\mathbf{x}$ and/or the decisions $\mathbf{a}$ may be categorical values, *e.g.*, in recommender systems, the input may be a user ID and the decision may be an item ID, both of which are categorical values. The standard one-hot encoding is not a good representation for GP. Instead, we embed each unique ID as a latent variable in a low dimensional space, *e.g.*, $\mathbf{a}_k \in \mathbb{R}^Q, \mathbf{a}_k \sim \mathcal{N}(0, \mathbf{I})$. This approach is closely related to variational multi-output GPs [11]. Deep GPs [12, 13] can be considered if the distribution of immediate feedback is heavily non-stationary.

With a surrogate model for the immediate feedback, we have all the pieces to estimate the accumulative metric from (2). The integral is generally intractable but could be approximated by the methods like Monte Carlo sampling. Note that the resulting quantity $v(\mathbf{M}_i)$ is deterministic as all the involved probability distributions are integrated out. It can serve as an estimator for the accumulative metric but is *unable* to be used for exploration-exploitation tradeoff. In order to construct an efficient MSPS method, we need to represent the accumulative metric as a random variable, of which the uncertainty reflects the current belief of its value according to the surrogate model, which is often referred to as *model uncertainty*.

### 3.2  Estimation of the accumulative metric

To derive the accumulative metric as a random variable that reflects model uncertainty, we first need to remove the uncertainty from the noise distribution, which corresponds to *aleatoric uncertainty*. This is particularly crucial for the case of a binary immediate feedback, which will be explained in the next section. Firstly, we derive the expected immediate feedback from the noise distribution, i.e., $\bar{m} = \langle m \rangle_{p(m|f)}$. In the case of a normal noise distribution, the expected immediate feedback is the mean of the noise distribution, $\bar{m} = \int m \mathcal{N}(m; f, \sigma^2) \, \mathrm{d}m = f$. Then, we derive the predictive expected immediate feedback from a inferred GP surrogate model by a change of random variable,

$p(\bar{\mathbf{m}}|\mathbf{A}, \mathbf{X}, \mathcal{D}) = p(\mathbf{f}|_{\mathbf{f}=\bar{\mathbf{m}}}|\mathbf{A}, \mathbf{X}, \mathcal{D})$, where $p(\mathbf{f}|\mathbf{A}, \mathbf{X}, \mathcal{D})$ is the noise-free predictive distribution from GP conditioned on the collected data via online experiments.

Consider a list of inputs $\mathbf{X} = (\mathbf{x}_1, \ldots, \mathbf{x}_T)$ and the decision space $\mathcal{A}$ being discrete, denoted as $\mathbf{A} = (\mathbf{a}_1, \ldots, \mathbf{a}_K)$. Given a model $\mathbf{M}_i$, the distribution of the model can be represented as a matrix $\mathbf{P} \in [0,1]^{K \times T}$, where each entry $p_{ij} = p(\mathbf{a}_i|\mathbf{x}_j)$. The accumulative metric is defined as the sum of immediate feedback weighted by the inputs and decision probabilities. This allows us to derive the accumulative metric as a random variable $\hat{v}|\mathbf{M}_i, \mathcal{D}$,

$$\hat{v}|\mathbf{M}_i, \mathcal{D} = \frac{1}{T}\mathbf{P}_:^\top \bar{\mathbf{m}}, \quad \bar{\mathbf{m}} \sim p(\bar{\mathbf{m}}|\mathbf{A}, \mathbf{X}, \mathcal{D}), \tag{4}$$

where the subscript : denotes the vectorization of a matrix and $\bar{\mathbf{m}}$ is the vector of expected immediate feedback corresponding to the combinatorial of $\mathbf{X}$ and $\mathbf{A}$, denoted as $\mathbf{W} = ((\mathbf{a}_1, \mathbf{x}_1), \ldots, (\mathbf{a}_K, \mathbf{x}_1), \ldots, (\mathbf{a}_K, \mathbf{x}_T))$. As the change of random variable in (4) is a linear operation, the resulting random variable $\hat{v}$ is jointly GP distributed with $\bar{\mathbf{m}}$. It turns out that the resulting distribution $p(\hat{v}|\mathbf{M}_i, \mathcal{D})$ can be derived in closed-form,

$$p(\hat{v}|\mathbf{M}_i, \mathcal{D}) = \mathcal{N}\left(\frac{1}{T}\mathbf{P}_:^\top \mathbf{K}_* \mathbf{K}^{-1} \mathbf{m}, \frac{1}{T}\mathbf{P}_:^\top (\mathbf{K}_{**} - \mathbf{K}_* \mathbf{K}^{-1} \mathbf{K}_*^\top)\mathbf{P}_:\right), \tag{5}$$

where $\mathbf{m}$ is the recorded immediate feedback in $\mathcal{D}$, $\mathbf{K}$ is the covariance matrix among the observed data $\mathcal{D}$, $\mathbf{K}_*$ is the cross-covariance matrix between $\mathbf{W}$ and $\mathcal{D}$ and $\mathbf{K}_{**}$ is the covariance among $\mathbf{W}$.

Note that the expectation of the random variable $\hat{v}$ recovers the accumulative metric estimator in (2), *i.e.*, $v(\mathbf{M}_i) = \int \hat{v} p(\hat{v}|\mathbf{M}_i, \mathcal{D}) \, \mathrm{d}\hat{v}$. As the probability distributions of inputs and decisions are represented in the matrix $\mathbf{P}$ and the uncertainty from the noise distribution is removed, the uncertainty in $\hat{v}$ is a result of the model uncertainty of the GP surrogate model, which is crucial for the exploration-exploitation tradeoff.

For a real world problem, $\mathcal{D}$ often contains lots of data points, for which the cubic complexity of exact GP inference is too expensive. For scalability, we use the variational sparse GP approximation [14]. It augments the original data with a set of pseudo data $\mathbf{u}$ at the corresponding locations $\mathbf{Z}$. Such an augmentation does not change the original model distribution $p(\mathbf{f}|\mathbf{A}, \mathbf{X}) = \int p(\mathbf{f}|\mathbf{u}, \mathbf{A}, \mathbf{X}, \mathbf{Z})p(\mathbf{u}|\mathbf{Z}) \, \mathrm{d}\mathbf{u}$. With an efficient variational lower bound, the computational complexity reduces from $O(N^3)$ to $O(NC^2)$, where $C$ is the number of pseudo data. The inference result of sparse GP is often represented by the variational posterior of the pseudo data, denoted as $q(\mathbf{u}) = \mathcal{N}(\mathbf{m}_\mathbf{u}, \mathbf{S}_\mathbf{u})$. With sparse GP approximation, the distribution $p(\hat{v}|\mathbf{M}_i, \mathcal{D})$ becomes

$$p(\hat{v}|\mathbf{M}_i, \mathcal{D}) = \mathcal{N}\left(\frac{1}{T}\mathbf{P}_:^\top \mathbf{K}_{*\mathbf{u}} \mathbf{K}_{\mathbf{uu}}^{-1} \mathbf{m}_\mathbf{u}, \frac{1}{T}\mathbf{P}_:^\top (\mathbf{K}_{**} - \mathbf{K}_{*\mathbf{u}}(\mathbf{K}_{\mathbf{uu}}^{-1} - \mathbf{K}_{\mathbf{uu}}^{-1}\mathbf{S}_\mathbf{u}\mathbf{K}_{\mathbf{uu}}^{-1})\mathbf{K}_{*\mathbf{u}}^\top)\mathbf{P}_:\right), \tag{6}$$

where $\mathbf{K}_{\mathbf{uu}}$ is the covariance matrix among the pseudo data and $\mathbf{K}_{*\mathbf{u}}$ is the cross-covariance matrix between $\mathbf{W}$ and the pseudo data.

For a large problem, the variance calculation in (6) can also be very expensive as $\mathbf{K}_{**}$ is a $KT$-by-$KT$ matrix. For efficient computation, we use a FITC approximation [15] at prediction time, *i.e.*, $p_{\text{FITC}}(\mathbf{f}|\mathbf{u}, \mathbf{A}, \mathbf{X}, \mathbf{Z}) = \mathcal{N}(\mathbf{K}_{\mathbf{fu}}\mathbf{K}_{\mathbf{uu}}^{-1}\mathbf{u}, \mathbf{\Lambda})$, where $\mathbf{\Lambda} = \text{diag}\left(\mathbf{K}_{\mathbf{ff}} - \mathbf{K}_{\mathbf{fu}}\mathbf{K}_{\mathbf{uu}}^{-1}\mathbf{K}_{\mathbf{fu}}^\top\right)$ and $\text{diag}(\cdot)$ makes a matrix into a diagonal matrix by letting off-diagonal entries be zero. Note that, although the conditional distribution $p(\mathbf{f}|\mathbf{u})$ is independent among the entries of $\mathbf{f}$, the resulting distribution $p(\bar{\mathbf{m}}|\mathbf{A}, \mathbf{X}, \mathcal{D})$ is still correlated due to the correlation from the pseudo data. With the FITC approximation, the mean $p(\hat{v}|\mathbf{M}_i, \mathcal{D})$ remains to be the same, while the variance becomes $\frac{1}{T}\mathbf{P}_:^\top (\mathbf{\Lambda} + \mathbf{K}_{*\mathbf{u}}\mathbf{K}_{\mathbf{uu}}^{-1}\mathbf{S}_\mathbf{u}\mathbf{K}_{\mathbf{uu}}^{-1}\mathbf{K}_{*\mathbf{u}}^\top)\mathbf{P}_:$, in which only the diagonal entries of $\mathbf{K}_{**}$ needs to be computed.

### 3.3 Binary immediate feedback

In industrial use cases, binary immediate feedback is widely used because it is easy to calculate and easy to interpret by human, *e.g.*, whether a user has responded to a shown item, whether a customer has purchased an item or whether a user has played a music or a movie. To apply our method to binary immediate feedback, we need to modify the GP surrogate model.

Firstly, we need to change the noise distribution to a Bernoulli distribution, $p(m|f) = \sigma(f)^m(1 - \sigma(f))^{1-m}$, where $\sigma(\cdot)$ is a link function that squashes the value of $f$ to be in $(0, 1)$. The most

common link function is the logistic function. This makes the GP surrogate model become a GP binary classification model, of which the marginal likelihood is no longer closed-form. For both tractability and scalability, we use stochastic variational sparse GP approximation [16], of which the intractable 1D integral in the variational lower bound is approximated by Gauss–Hermite quadrature. Then, we derive the expected immediate feedback from the Bernoulli distribution, which is the probability of the immediate feedback being one, $i.e.$, $\bar{m} = \sum_{m \in \{0,1\}} m\, p(m|f) = \sigma(f)$. The predictive expected immediate feedback from a inferred GP surrogate model can be derived by a change of random variable,

$$p(\bar{\mathbf{m}}|\mathbf{A}, \mathbf{X}, \mathcal{D}) = p(\mathbf{f}|_{\mathbf{f}=\sigma^{-1}(\bar{\mathbf{m}})}|\mathbf{A}, \mathbf{X}, \mathcal{D}) \left| \frac{\mathrm{d}\sigma^{-1}(\bar{\mathbf{m}})}{\mathrm{d}\bar{\mathbf{m}}} \right|. \tag{7}$$

where $\sigma^{-1}(\cdot)$ is the inverse of the link function. Both $\sigma(\cdot)$ and $\sigma^{-1}(\cdot)$ are scalar functions. We use $\sigma^{-1}(\bar{\mathbf{m}})$ to denote applying $\sigma^{-1}(\cdot)$ to the individual entries of $\bar{\mathbf{m}}$. For binary immediate feedback, the random variable of the accumulative metric $\hat{v}$ defined in (4) no longer has a closed form probability density function. Fortunately, we can efficiently sample from $p(\hat{v}|\mathbf{M}_i, \mathcal{D})$, by first drawing a sample from the "noise-free" GP surrogate model and compute the sample of $\hat{v}$ according to (4), $i.e.$,

$$\hat{v}_i = \frac{1}{T}\mathbf{P}_{:}^{\top}\sigma(\mathbf{f}_i), \quad \mathbf{f}_i \sim p(\mathbf{f}|\mathbf{A}, \mathbf{X}, \mathcal{D}). \tag{8}$$

Note that for binary immediate feedback, it is crucial to derive the random variable $\hat{v}$ from the expected immediate feedback $\bar{m}$ instead of the original immediate feedback $m$. Imagine that we derive $\hat{v}$ from $m$ by replacing $\bar{\mathbf{m}}$ with $\mathbf{m}$. The consequence is that the variance of $\hat{v}$ will be at maximum if the expected immediate feedback $\mathbf{m}$ equals to 0.5, no matter how small the model uncertainty is. In this case, the uncertain in $\hat{v}$ does not reflect the amount of unknowns in the surrogate model. Instead, deriving $\hat{v}$ from $\bar{m}$ can avoid this problem because the uncertainty from the noise distribution is excluded.

## 4  Choosing the next online experiment

After deriving the probability distribution of the accumulative metric, we use an acquisition function to guide the choice of the next online experiment. We consider the acquisition functions $\alpha(\cdot)$ widely used in BO $e.g.$ expected improvement (EI), probability of improvement (PI) and upper confidence bound (UCB). A major difference to BO is that the space of choices is no longer the same as the input space of the surrogate model. In our case, the space of choices are the set of candidate models, while the input to the surrogate model is the input to the ML model and the decision from the ML model. As a result, the acquisition functions designed by considering an extra hypothetical evaluation such as entropy search [17] cannot be used within our method. For binary feedback, the acquisition functions are not closed form. We use Monte Carlo sampling to compute the acquisition function by drawing samples from the distribution of the accumulative metric.

We name the resulting algorithm as automated online experimentation (AOE), of which the overall procedure is shown in Algorithm 1. We start with an initial dataset $\mathcal{D}_0$, which may be collected by deploying the model online. The model can be randomly chosen or chosen according to some domain knowledge or offline accuracy measure. Note that often the training data for the candidate models may be used to train the surrogate model as well. At each iteration, we first update surrogate model by inferring the variational posterior distribution as mentioned in Section 3. Then, we score all the candidate models with the acquisition function, which takes as inputs the distribution of the accumulative metric and select the model with the highest score. The selected model is deployed online for data collection. The collected data are augmented into the dataset for updating the surrogate model. We repeat this process until the online experiment budget is over. Then, the best model can be estimated from the latest surrogate model.

## 5  Related Work

Model selection [18] is a classical topic in ML. The standard paradigm of model selection considers a model in insolation and aims at selecting a model that has the best predictive power for unseen data based on an offline dataset. Common techniques such as cross-validation, bootstrapping, Akaike information criterion [AIC, 19] and Bayesian information criterion [BIC, 20] have been widely used

**Algorithm 1:** model selection with automated online experiments (AOE)

---

**Result:** Return the ML system with the highest accumulative metric
Collect the initial data $\mathcal{D}_0$;
**while** *Online experiment budget is not over* **do**

   Infer $p(\mathbf{f}|\mathbf{A}, \mathbf{X}, \mathcal{D}_{t-1})$ with VI on surrogate model ;
   Identify $\mathbf{M}_t = \arg\max_{\mathbf{M}_i \in \mathcal{M}} \alpha(\mathbf{M}_i)$;
   Deploy $\mathbf{M}_t$ and construct $\mathcal{D}_t$ by augmenting the collected data into $\mathcal{D}_{t-1}$ ;
**end**

---

for scoring a model's predictive power based on a given dataset. As scoring all the candidate models does not scale for complex problems, many recent works focus on tackling the problem of searching a large continuous and/or combinatorial space of model configurations, ranging from hyper-parameter optimization [HPO, 10, 21], automatic statistician [22, 23, 24, 25] to neural network architecture search [NAS, 26]. A more recent work [27] jointly considers the scoring and searching problem for computational efficiency. Online model selection [28, 29] is an extension of the standard model selection paradigm. It still treats a model in isolation but considers the online learning scenario, in which data arrive sequentially and the models are continuously updated. This is different to MSPS, which views a model in the context of a bigger system and actively controls the data collection.

In reinforcement learning (RL), a model is considered as a decision mechanism, referred to as a *policy*, and is evaluated for its associated accumulative rewards. Off-policy evaluation (OPE) [4, 5, 6, 7, 8, 9] predicts the value of a new policy from an offline dataset logged by another policy. The definition of the value of a policy shares a similar format with the accumulative metric, which allows us to develop some baseline methods based on OPE. The major difference is that our method considers data collection as part of the selection process, which leads to the derivation of the probability distribution of the accumulative metric instead of an estimator of the value of a policy as in OPE. Many works in RL also exploit the idea of Bayesian modeling and BO [30, 31], where GP is used to model the value function and obtain policy gradient via Bayesian quadrature. Lee et al. [32] use Bayesian models to represent the belief distribution over latent state space and perform off-policy update via a trust region method. Recently, Letham and Bakshy [33] use multi-fidelity BO for policy search by correlating online and offline metrics. Their approach relies on the assumption that there exists an offline metric correlated with the online metric and suffers from the limitation of BO, which requires the search space to be relatively low-dimensional. Russo [34] considers the best-arm selection problem in contextual bandits, of which the used techniques are related to model selection but operate at a lower granularity.

Optimal experimental design is an area of research that focuses on techniques for efficient usage of limited resources in training models and data collections. Bayesian optimal experimental design tackles this problem by constructing a predictive model for possible experimental outcome, and seeks to optimize the expected information gain based on the posterior predictive estimation [35, 36, 37]. Rather than selecting model based on logging data, they take a parallel approach of optimizing data selection process based on the logging information, which has been successfully applied to various settings including bioinformatics [38], active learning [39], and neuroscience [40].

# 6 Experiment

We demonstrate the performance of AOE on automating online experiments for model selection. We construct two simulators based on real data to perform the evaluation since evaluation on a production system is not reproducible. We compare AOE with five baseline methods: (i) directly applying BO on the collected accumulative metrics; (ii) using two OPE methods (importance sampling (IS) and doubly robust (DR)) to estimate the accumulative metrics at each iteration and greedily deploy the model with the best estimated metric, denoted as IS-g and DR-g respectively; and finally (iii) using two OPE methods to estimate the accumulative metrics with their empirical variance and choose the model to deploy according to an acquisition function (EI), denoted as IS-EI and DR-EI, respectively. For the details of the baseline methods see the supplement material.

**Classification.** Inspired by how OPE works, we use a classification dataset to construct the first simulator. We consider the candidate models to be multi-class classifiers, and when deployed online,

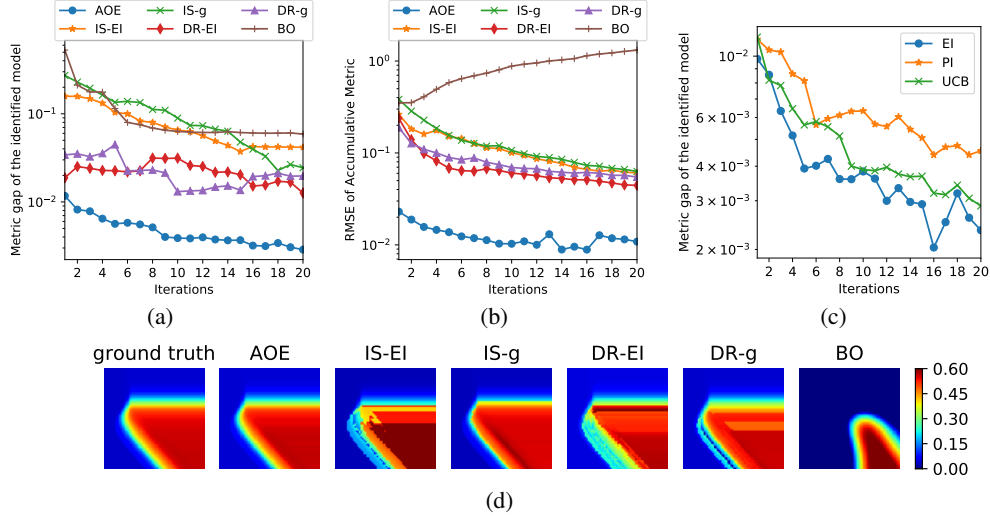

Figure 1: Results of the classification experiment. (a) Comparison of AOE and five baseline methods through the 20 sequential online experiments (refer to as iterations). The $y$-axis shows the gap in the accumulative metric between the optimal model and the estimated best model by each method. The average metric gaps after Iteration 20 are **0.0029**, 0.042, 0.024, 0.013, 0.020, 0.059 for **AOE**, IS-EI, IS-g, DR-EI, DR-g, BO respectively. (b) RMSE of the estimated accumulative metrics for all the candidate models from each method. It is calculated from the same set of experiments as the ones in (a). The average RMSE after Iteration 20 are **0.011**, 0.061, 0.063, 0.044, 0.054, 1.31 for for **AOE**, IS-EI, IS-g, DR-EI, DR-g, BO respectively. (c) Comparison of AOE using different acquisition functions. (d) Heat maps of the estimated accumulative metric of all the candidate models after Iteration 20 comparing with the ground truth.

a model will be given a set of inputs but only receives binary feedback about whether the predicted class is correct. The task is to identify the model with the best accumulative metric in the smallest number of deployments, where the accumulative metric is the average accuracy on the hold-on set in this case. We use the "letter" dataset from UCI repository [41] and randomly take 200 data points for training and use the rest for "online" experiments. In each online experiment, we randomly select 200 data points and pass them to the "deployed" model and record the binary feedback and accumulative metric. We generate the set of candidate models by changing the two tuning parameter of support vector machine (SVM), $C$ and $\gamma$. The candidate model set is generated on a 100x100 grid in the space of the two parameters in log scale. In order to compare with the OPE-based baselines, all the decision **a** is augmented with a $\epsilon$-greedy step with $\epsilon = 0.05$. We use a GP binary classifier with Matérn $\frac{3}{2}$ kernel as the surrogate model, using 2000 inducing points. We use EI as the acquisition function implemented in GPyOpt [42]. Each run consists of 20 sequential online experiments with the first deployed model randomly picked. Each method repeatedly runs 20 times.

Figure 1a shows the comparison of all the methods as an average of 20 repeated runs. The performance at each iteration is measured as the gap in the accumulative metric between the optimal model and the estimated best model. The models picked by AOE at all the iterations have significantly smaller metric gaps and the average metric gaps after Iteration 20 are **0.0029**, 0.042, 0.024, 0.013, 0.020, 0.059 for **AOE**, IS-EI, IS-g, DR-EI, DR-g, BO respectively. BO stops improving after about Iteration 10 because it can only use the recorded accumulative metrics. DR-g and DR-EI are better than IS-g and IS-EI due to their lower variance estimator. Figure 1b shows the rooted mean square error (RMSE) between the estimated accumulative metrics for all the candidate models from each method, which is averaged across 20 runs. The average RMSE after Iteration 20 are **0.011**, 0.061, 0.063, 0.044, 0.054, 1.31 for for **AOE**, IS-EI, IS-g, DR-EI, DR-g, BO respectively. The RMSE of BO does not decrease due to the wrong generation from a few data points, which is worse than a flat prediction in the beginning. Figure 1c compares the different choices of acquisition functions (EI, PI, UCB) for AOE, all of which performs similarly with EI being slightly better. Figure 1d shows the heat map visualization of the estimated accumulative metrics of all the candidate models after Iteration 20 from one of the 20 runs comparing with the ground truth. The $x$- and $y$-axis of the heat maps correspond

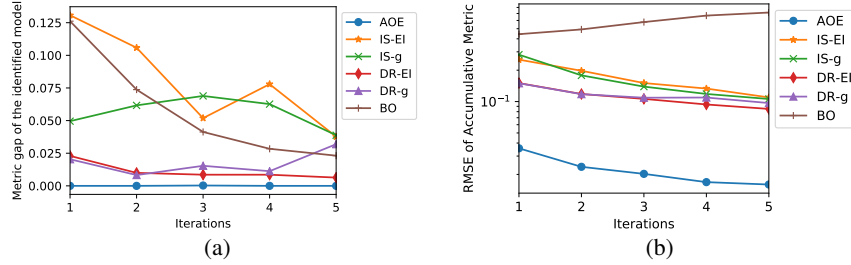

Figure 2: Results of the recommender system experiment. (a) Comparison of AOE and five baseline methods through the five sequential online experiments. The $y$-axis shows the gap in the accumulative metric between the optimal model and the estimated best model by each method. The average metric gaps after Iteration 5 are **0.**, 0.038, 0.039, 0.0063, 0.032, 0.023 for **AOE**, IS-EI, IS-g, DR-EI, DR-g, BO respectively. (b) RMSE of the estimated accumulative metrics of all the candidate models by each method. The average RMSE after Iteration 5 are **0.016**, 0.11, 0.11, 0.085, 0.097, 0.715 for for **AOE**, IS-EI, IS-g, DR-EI, DR-g, BO respectively.

to the two parameters $C$ and $\gamma$ in log scale. AOE has the best visual resemblance to the ground truth among others, which is consistent with the RMSE result. See the supplement for the details.

**Recommender System.** We consider model selection for recommender system, which aims to select the best recommender based on its online performance. In this experiment, a recommender system takes a user ID as input and returns an item ID for recommendation. For each recommendation, it receives binary feedback indicating whether the user has responded to the recommended item. The performance of a recommender system is measured by the average response rate, which is the accumulative metric. We use the MovieLens 100k data [43] to construct the simulator for online experiments. Binary feedback is simulated according to the response probability that is computed by filling all the missing entries in the rating data with zero and mapping a 0-5 rating evenly into a probability between $[0.05, 0.95]$. We randomly take 20% data for training and trained ten models using the Surprise package [44]. In an online experiment, given a user ID, a trained model returns a predicted response probability and the recommendation is generated by taking the top five items and uniformly choosing one from them. The recommendation is augmented with $\epsilon$-greedy, where $\epsilon = 0.05$. Each run consists of five sequential online experiments. In each online experiment, every user is recommended five times.

Figure 2 shows the comparison with the average of 20 repeated runs. As shown in Figure 2a, AOE always identifies the best model from the first iteration and BO fails to reduce the RMSE because the surrogate model of BO is not aware of the objective ranged between zero and one. The average metric gaps after Iteration 5 are **0.**, 0.038, 0.039, 0.0063, 0.032, 0.023 for **AOE**, IS-EI, IS-g, DR-EI, DR-g, BO respectively. In Figure 2b, we show the RMSE of the estimated accumulative metrics. The RMSE from AOE continuously decreases, while it always correctly identifies the best model. The average RMSE after Iteration 5 are **0.016**, 0.11, 0.11, 0.085, 0.097, 0.715 for for **AOE**, IS-EI, IS-g, DR-EI, DR-g, BO respectively.

# 7 Discussion

The model selection for production system does not fit into the classical model selection paradigm. We propose a new approach by taking data collection into the model selection process and selecting the best model via iterative online experiments. It allows selection from a much larger pool of candidates than using A/B testing and gives more accurate selection than off-policy evaluation by actively reducing selection bias. We design a GP surrogate model for predicting immediate feedback and derive the distribution of the accumulative metric. The model to deploy at each iteration is picked by balancing the predicted accumulative metric and the uncertainty of the prediction due to limited data. With simulated experiments from real data, we show that AOE performs significantly better than all the baselines in terms of identifying the best model and estimating the accumulative metric.

The concept of iterative model deployment also appear in bandit algorithms. A bandit algorithm performs exploration-exploitation for individual user interactions and continuously updates the model.

A major difference to our paradigm is that a bandit algorithm is often applied to the decision making scenarios that have tight time constraints because of its short decision time, in which the model is either not updated or updated with incremental learning after each action, while our method selects among different candidate models and each online experiment contains lots of user interactions, which allows us to consider more expensive surrogate models and retrain the surrogate model after each action. RL as a broader framework also considers the problem about evaluating and updating a policy from recorded data with respect to a generic form of reward, which may be delayed and depend on sequential actions. MSPS can be viewed as a special case of RL, in which the reward is an average of immediate feedback. This special setting allows us to develop a dedicated surrogate model and make efficient use of data, which is not applicable to the generic RL setting. On the other hand, RL offers an interesting future direction for handling broader types of accumulative metric beyond the form of average.

## Broader Impact

In this paper, the authors present a new framework of model selection for production system (MSPS), in which the data collection from automated online experiments is used as part of the model selection procedure. In particular, we develop AOE, a MSPS method that iteratively select models to be deployed online and identify the model with the highest metric of interest from a large pool of candidate models in a few number of online deployments.

AOE could be applied to improve the quality of the model selection process for industrial ML service development. This type of methods could be implemented either as a part of the in-house development toolset of individual companies or as a component of automated ML service on cloud platforms. The adoption of such tooling could increase the development speed of industrial ML applications and provide better understanding and control of the release of new features and improvement before large scale deployment. With a more accurate prediction of the online metric of a system improvement, ML developers can better identify the impactful system improvements and focus the development effort on them. It also can let the development team and a wider part of a company have a clear picture of the potential impact and limitation of a project before the development has finished.

The automated ML model selection, update, deployment tools including AOE tend to focus on a single metric for mathematical convenience, but the social impact of a ML system such as diversity, fairness is hard to summarize into a single metric. The adoption of such tooling without careful consideration can result into overly optimize for the single metric and being blind about broad social impacts, which can potentially lead to undesirable outcomes. The research about understanding and constraining automated algorithm decisions with respect to its wider impact, *e.g.*, safe reinforcement learning, is very important and could mitigate the risk of causing harmful consequences.

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
