[Supplementary Material]

# Supplemental Material for "Model Selection for Production System via Automated Online Experiments"

## A Experiment Details

In the following section, we will present the additional details about our experiments that do not fit in the main text.

### A.1 Baseline methods

As MSPS is a new framework for model selection, we construct five baseline methods by extending the related method into our scenario and compare with AOE. The five baseline methods are as follows:

**BO** For each online experiment, we can have an unbiased estimate of the accumulative metric under the deployed model as mentioned in Section 2. We directly apply Bayesian Optimization (BO) to the model selection problem by taking the set of candidate models as the input space and the estimate of the accumulative metric from online experiments as the output and treating MSPS as an optimization problem. We use the default setting of BO in GPyOpt, where the surrogate model is a Gaussian process (GP) regression model with a Gaussian noise distribution and a Mátern 5/2 kernel. Expected Improvement (EI) is used as the acquisition function. For the classification experiment, as the candidate models are naturally generated from a 2D space of the SVM parameters C and $\gamma$, we use the values of these two parameters to identify individual candidate models and use this 2D space as the search space for BO. However, for the recommender system experiment, there are no natural representations for the candidate models. We treat each candidate model as a categorical value, which leads to its bad performance.

**IS-g / DR-g** Off-policy evaluation (OPE) methods can provide an estimate of the accumulative metric. We use two popular OPE methods, importance sampling (IS) and doubly robust (DR) to estimate the accumulative metric after each online experiment and greedily choose the candidate model with the highest estimated accumulative metric for the next online experiment. We denote the resulting two methods as IS-g and DR-g respectively.

**IS-EI / DR-EI** IS-g and DR-g suffer from the fact that there is no exploration mechanism. To offer a stronger baseline, we not only use IS and DR to estimate the accumulative metric, but also calculate the empirical variance of the resulting estimate. Then, we score the candidate models according to an acquisition function (EI is used in the experiments) and select the next model to deploy with the highest score. The resulting methods are denoted as IS-EI and DR-EI respectively.

As there are limited information to be gained by repeatedly deploying the same model online, we exclude the models that have been deployed when choosing the next model to deploy for all the methods including AOE.

### A.2 Classification

We take the inspiration from the OPE literature and construct an online experiment scenario using a classification dataset. We simulate the "online" deployment scenario as follows: a multi-class classifier is given a set of inputs; for each input, the classifier returns a prediction of the label and only a binary immediate feedback about whether the predicted class is correct is available. The performance of a classifier is measured by the average accuracy on the hold-out set, which corresponds to the accumulative metric. As only one model can be deployed at a time and in each deployment a small subset of the hold-out data are used, the task is to select the best-performing model in the smallest number of deployments.

We use the "letter" dataset from UCI repository [41]. There are, in total, 20,000 data points in the dataset. We randomly sample 200 data points for training and use the rest for "online" experiments. In each online experiment, we randomly select 200 data points from the hold-out set and pass them to the "deployed" model and record the binary feedback and accumulative metric.

Figure 3: Additional results of the classification experiment. (a) Comparison of AOE and five baseline methods through the 20 sequential online experiments (refer to as iterations). The $y$-axis shows the gap in the accumulative metric between the optimal model and the estimated best model by each method. (b) RMSE of the estimated accumulative metrics for all the candidate models from each method. The error bars in both (a) and (b) indicate the confidence interval of the estimated mean by two times of the standard deviation.

Table 1: The metric gap and RMSE after Iteration 20 in the classification experiment

|        | Metric Gap | RMSE |
|--------|------------|------|
| AOE    | **0.0029 (0.0033)** | **0.011 (0.0053)** |
| IS-EI  | 0.042 (0.042) | 0.061 (0.019) |
| IS-g   | 0.024 (0.036) | 0.063 (0.027) |
| DR-EI  | 0.013 (0.016) | 0.044 (0.027) |
| DR-g   | 0.020 (0.024) | 0.054 (0.026) |
| BO     | 0.059 (0.15) | 1.31 (0.54) |

We consider support vector machine (SVM) as the multi-class classifier and generate the set of candidate models by varying the two tuning parameters of SVM for training, $C$ and $\gamma$. To demonstrate that AOE can select a good model from a large set of candidates, we generate in total 10,000 candidate models by choosing $C$, and $\gamma$ from a 100x100 grid in the space of these two parameters in log scale. We follow the guideline from previous works and consider $C$ between $2^{-5}$ and $2^{15}$ and $\gamma$ between $2^{-15}$ and $2^3$. In order to compare with the OPE-based baselines, all the predictions from an SVM are augmented with a $\epsilon$-greedy step, *i.e.*, the predicted label is sampled according to a categorical distribution, in which the class predicted by the SVM has $1 - \epsilon$ probability and the rest classes evenly share the probability $\epsilon$. We set $\epsilon = 0.05$.

We use a GP binary classifier with a Matérn $\frac{3}{2}$ kernel as the surrogate model. We use 2000 inducing points and EI as the acquisition function. When training the surrogate model, we use Adam as the gradient optimizer for variational inference, which runs for 600 epochs with the mini-batch size being 100 and the learning rate being 0.001 with stratified sampling. We use logistic regression as the baseline model for DR-based baselines.

Each model selection experiment consists of 20 sequential online experiments and the model deployed in the first experiments is randomly picked according to a uniform distribution for all the methods. We repeatedly run 20 experiments for each model. In each repeated run, the set of candidate models are the same but the first deployed model and the data points sampled for each online experiment may be different due to random sampling.

Figure 4: Additional results about the comparison of acquisition functions. It compares the performance of different acquisition function used by AOE in the classification experiment. (a) Comparison of different acquisition functions in terms of the gap in the accumulative metric between the optimal model and the estimated best model. (b) Comparison of different acquisition function in terms of RMSE of the estimated accumulative metrics. The error bars in both (a) and (b) indicate the confidence interval of the estimated mean by two times of the standard deviation.

Table 2: The metric gap and RMSE after Iteration 5 in the recommender system experiment

|        | Metric Gap       | RMSE            |
|--------|------------------|-----------------|
| AOE    | **0. (0.)**      | **0.016 (0.0029)** |
| IS-EI  | 0.038 (0.043)    | 0.11 (0.031)    |
| IS-g   | 0.039 (0.048)    | 0.11 (0.043)    |
| DR-EI  | 0.0063 (0.024)   | 0.085 (0.018)   |
| DR-g   | 0.032 (0.039)    | 0.097 (0.027)   |
| BO     | 0.023 (0.043)    | 0.715 (0.062)   |

Figure 3a shows the comparison of all the methods with error bars in terms of the the gap in the accumulative metric between the optimal model and the estimated best model. Figure 3b shows the average rooted mean square error (RMSE) of the estimated accumulative metrics after each iteration with error bars. The error bars indicate the confidence interval of the estimated mean by two times of the standard deviation. The average metric gaps and average RMSE of all the methods after Iteration 20 are shown in Table 1. The values in the parentheses indicates the standard deviation of the metric gap and RMSE across the 20 repeated runs.

Apart from the comparison between AOE and the baseline methods. We also compare the performance of AOE when using different acquisition functions. Figure 4a shows the comparison of different acquisition functions with error bars in terms of the the gap in the accumulative metric between the optimal model and the estimated best model. Figure 4b shows the average RMSE of the estimated accumulative metrics after each iteration with error bars with different acquisition functions. The error bars indicate the confidence interval of the estimated mean by two times of the standard deviation.

To illustrate the behaviors of AOE and the baseline methods during the model selection process, we visualize the mean and standard deviation of the estimated accumulative metrics after Iteration 1, 5, 10, 15, and 20 from individual methods in Figure 7. The visualization uses one of the 20 runs. Note that, to provide more information, Figure 1d and Figure 7 use different runs.

## A.3 Recommender System

We demonstrate AOE on the problem of model selection for recommender system, which aims to select the best recommender based on its online performance. In this experiment, we consider that

Figure 5: Additional results of the recommender system experiment. (a) Comparison of AOE and five baseline methods through the 20 sequential online experiments (refer to as iterations). The $y$-axis shows the gap in the accumulative metric between the optimal model and the estimated best model by each method. (b) RMSE of the estimated accumulative metrics for all the candidate models from each method. The error bars in both (a) and (b) indicate the confidence interval of the estimated mean by two times of the standard deviation.

a recommender system takes a user ID as input and returns an item ID for recommendation. For each recommendation, it receives binary feedback indicating whether the user has responded to the recommended item. We measure the performance of such a recommender system by the average response rate, which corresponds to the accumulative metric. We construct a simulator by using the MovieLens 100k data [43]. Given a user ID and an item ID, the binary feedback is simulated by drawing a sample from a Bernoulli distribution, in which the probably of being one is specified by the response probability corresponding to the user ID and item ID pair. The MovieLens 100k data provide the ratings corresponding to a list of user and item pairs. We filter items that have average rating below a threshold of 0.2. The ratings range between one and five. We generate a full table of the response probability for all the user and item combinations by first filling all the missing entries in the rating data with zero and mapping the resulting 0-5 rating evenly to a probably between $[0.05, 0.95]$, *i.e.*, 0.05 for 0, 0.23 for 1, 0.41 for 2, 0.59 for 3, 0.77 for 4 and 0.95 for 5.

We do not use any user and item features and randomly take 20% of the entries in the response probability table for training the candidate models. We trained ten models using the Surprise package [44] with their default setting. The full list of the names of the models are SVD, BaselineOnly, Co-Clustering, KNNBaseline, KNNWithMeans, NormalPredictor, NMF, KNNWithZScore, KNNBasic, SlopeOne. At prediction time, each of these models predicts a response probability given a pair of user ID and item ID. In an online experiment, given a user ID, a trained model predicts the response probabilities of all the items, and the recommendation is generated by taking the top five items and randomly choosing one from them following a uniform distribution. The recommendation is augmented with $\epsilon$-greedy, *i.e.*, the item for recommendation is sampled from a categorical distribution, in which the top five items have $(1 - \epsilon)/5$ probability and the rest items evenly share $\epsilon$ probability. We set $\epsilon = 0.05$.

The users and items are represented by their IDs, which are not good representations for GP. We augment a GP binary classifier by embedding the user and item IDs into two separate latent spaces as mentioned in Sec. 3.1 and use it as the surrogate model. We use 5D latent spaces for the user and item embedding separately and an RBF kernel. We use 1000 inducing points and EI as the acquisition function. When training the surrogate model, we use Adam as the gradient optimizer for variational inference, which runs for 200 epochs with the mini-batch size being 100 and the learning rate being

0.001. As the prediction task can also be viewed as matrix imputation, we use the KNNImputer from the scikit-learn package as the predictive model for DR-based baselines.

Each model selection experiment consists of five sequential online experiments and the model deployed in the first experiments is randomly picked according to a uniform distribution for all the methods. The data collected in each online experiment are generated by considering each user for recommendation five times. We repeatedly run 20 experiments for each model. In each repeated run, the set of candidate models is the same but the first deployed model and the data points sampled for each online experiment may be different.

Figure 5a shows the comparison of all the methods with error bars in terms of the gap in the accumulative metric between the optimal model and the estimated best model. Figure 5b shows the average RMSE of the estimated accumulative metrics after each iteration with error bars. The error bars indicate the confidence interval of the estimated mean by two times the standard deviation. The metric gaps and RMSE of all the methods after Iteration 5 are shown in Table 2. To illustrate the behaviors of AOE and the baseline methods during the model selection process, we visualize the estimated accumulative metric after each iteration comparing with the ground truth in Figure 6. The visualization uses one of the 20 runs.

## B   Details about Sparse GP and Variational Inference

For scalability, we use the variational sparse GP approximation to speed up the inference. Variational sparse GP augments the original data with a set of pseudo data $\mathbf{u}$ at the corresponding locations $\mathbf{Z}$, which shares the same GP as in the original model. The input locations of the pseudo data $\mathbf{Z}$ lie in the joint space of the action and input $\mathbf{a}$ and $\mathbf{x}$. The resulting model is a joint GP between the original data and the augmented data $p(\mathbf{f}, \mathbf{u}|\mathbf{A}, \mathbf{X}, \mathbf{Z})$, which can also be written as a product of the conditional distribution $p(\mathbf{f}|\mathbf{u}, \mathbf{A}, \mathbf{X}, \mathbf{Z})p(\mathbf{u}|\mathbf{Z})$. Note that such an augmentation does not change the original model,

$$p(\mathbf{f}|\mathbf{A}, \mathbf{X}) = \int p(\mathbf{f}|\mathbf{u}, \mathbf{A}, \mathbf{X}, \mathbf{Z})p(\mathbf{u}|\mathbf{Z})\,\mathrm{d}\mathbf{u}. \tag{9}$$

Both $\mathbf{f}$ and $\mathbf{u}$ are latent variables. Variational sparse GP assumes a specific variational posterior distribution $q(\mathbf{f}, \mathbf{u}) = p(\mathbf{f}|\mathbf{u})q(\mathbf{u})$, where $q(\mathbf{u}) = \mathcal{N}(\mathbf{m_u}, \mathbf{S_u})$ is a multi-variate normal distribution, of which the mean and covariance matrix are variational parameters. A variational lower bound can be derived with the above variational posterior, of which the computational complexity reduces from $O(N^3)$ to $O(NC^2)$, where $C$ is the number of pseudo data. More details about the variational approximation can be found in [14].

After inferring the variational posterior of the sparse GP, the distribution of the accumulative metric conditioned on the observed data can be derived based on the variational posterior of sparse GP,

$$p(\hat{v}|\mathbf{M}_i, \mathcal{D}) = \mathcal{N}\left(\frac{1}{T}\mathbf{P}_:^\top\mathbf{K}_{*\mathbf{u}}\mathbf{K}_{\mathbf{uu}}^{-1}\mathbf{m_u}, \frac{1}{T}\mathbf{P}_:^\top(\mathbf{K}_{**} - \mathbf{K}_{*\mathbf{u}}(\mathbf{K}_{\mathbf{uu}}^{-1} - \mathbf{K}_{\mathbf{uu}}^{-1}\mathbf{S_u}\mathbf{K}_{\mathbf{uu}}^{-1})\mathbf{K}_{*\mathbf{u}}^\top)\mathbf{P}_:\right), \tag{10}$$

where $\mathbf{K}_{\mathbf{uu}}$ is the covariance matrix among the pseudo data, $\mathbf{K}_{*\mathbf{u}}$ is the cross-covariance matrix between $\mathbf{W}$ and the pseudo data, and $\mathbf{m_u}$ and $\mathbf{S_u}$ are the inferred variational parameters in $q(\mathbf{u})$. With this above derived distribution of the accumulative metric, we can apply an acquisition function for selecting a candidate model.

For a large problem, the variance calculation in the above distribution can also be very expensive as $\mathbf{K}_{**}$ is a $KT$-by-$KT$ matrix. For efficient computation, we apply a FITC approximation [15] at prediction time,

$$p_{\mathrm{FITC}}(\mathbf{f}|\mathbf{u}, \mathbf{A}, \mathbf{X}, \mathbf{Z}) = \mathcal{N}(\mathbf{K}_{\mathbf{fu}}\mathbf{K}_{\mathbf{uu}}^{-1}\mathbf{u}, \mathbf{\Lambda}), \tag{11}$$

where $\mathbf{\Lambda} = \mathrm{diag}\left(\mathbf{K}_{\mathbf{ff}} - \mathbf{K}_{\mathbf{fu}}\mathbf{K}_{\mathbf{uu}}^{-1}\mathbf{K}_{\mathbf{fu}}^\top\right)$ and $\mathrm{diag}\left(\cdot\right)$ makes a matrix into a diagonal matrix by letting off-diagonal entries be zero. Note that, although the conditional distribution $p(\mathbf{f}|\mathbf{u})$ is independent among the entries of $\mathbf{f}$, the resulting distribution $p(\bar{\mathbf{m}}|\mathbf{A}, \mathbf{X}, \mathcal{D})$ is still correlated due to the correlation from the pseudo data. With the FITC approximation, the resulting distribution of the accumulative metric becomes

$$p(\hat{v}|\mathbf{M}_i, \mathcal{D}) = \mathcal{N}\left(\frac{1}{T}\mathbf{P}_:^\top\mathbf{K}_{*\mathbf{u}}\mathbf{K}_{\mathbf{uu}}^{-1}\mathbf{m_u}, \frac{1}{T}\mathbf{P}_:^\top(\mathbf{\Lambda} + \mathbf{K}_{*\mathbf{u}}\mathbf{K}_{\mathbf{uu}}^{-1}\mathbf{S_u}\mathbf{K}_{\mathbf{uu}}^{-1}\mathbf{K}_{*\mathbf{u}}^\top)\mathbf{P}_:\right), \tag{12}$$

where only the diagonal entries of $\mathbf{K}_{**}$ needs to be computed.

Figure 6: The bar plot of the estimated accumulative metrics of all the candidate models after each iteration comparing with ground truth (denoted as "gt"). The results come from one of the 20 repeated runs. The $y$-axis shows the accumulative metric. In the $x$-axis, each group of bars corresponds to a candidate model (there are ten in total.) and each color of bars corresponds to all the compared methods plus the ground truth.

Figure 7: The heat map visualization of the estimated accumulative metrics of all the candidate models after Iteration 1, 5, 10, 15 and 20. The results come from one of the 20 repeated runs. The $x$- and $y$-axis of the heat maps correspond to the two parameters $C$ and $\gamma$ in log scale. Each column corresponds to a method. "(mean)" indicates the visualization of the mean of the estimation and "(std)" indicates the visualization of the standard deviation estimated by each method.