[Reviews · NeurIPS 2020]

Review 1

Summary and Contributions: This paper is about applying GPs and Bayesian optimization for online model selection, for example, deploying new Ad models and choosing the one that maximizes click. The approaches used: sparse GPs, and variational inference, are not new. The novelty is in the model and the application. Experimental results show clear improvement in performance. The paper is very well written, and was a pleasure to read.

Strengths: This paper proposes a theoretically well grounded model for accumulative metrics, and uses sparse GPs for scalability when optimizing the metrics. The case of non-linear binary feedback from GPs is also considered (although details on inference for this case is missing). The proposed model seems adequate in modeling practical online optimization problems. The experiments seem to be complete. Results show clear improvement in performance compared to the baselines.

Weaknesses: The m weakness is that methods applied to optimize the proposed model: VI and sparse GPs are not new, making the paper relatively straightforward.

Correctness: The technical details and the empirical evaluations seem correct, and extensive.

Clarity: Here are some suggestions to improve the clarity: Section 3.2: Explicitly mention that the epistemic or model uncertainty is being modeled here. More details about sparse GPs and VI can be provided in the appendix. Section 3.3: It can help to discuss the inference procedure (perhaps in the appendix) for the binary feedback case. Experiments: More experimental details should be provided. For instance, how is the probability matrix computed?

Relation to Prior Work: Prior work has been adequately addressed.

Reproducibility: Yes

Additional Feedback: Overall this is a nice paper, however more details should be added to the paper for better readability. Questions: How is the probability matrix P computed? Stylistic comment: Line 284, 289, 315, 318: The values should accompany the plot and should be in the figure captions. Some typos: Line 107: 'radial' Line 191: 'Imagine' Line 266: 'how OPE works' ======= Post rebuttal: My views about this paper are still positive, so I am keeping my score.


Review 2

Summary and Contributions: This paper studied the problem of model selection via a Gaussian process. The authors proposed an automated online experimentation mechanism that can efficiently perform model selection from a large pool of models with a small number of online experiments.

Strengths: - The paper is well-written, well-structured and easy to follow. - Discussion on the differences among model selection algorithms, deep reinforcement algorithms, and bandit algorithms is provided. - Five baseline models are compared in the experiments to demonstrate the superiority of the proposed model selection algorithm.

Weaknesses: - Motivation of utilizing a Gaussian process but not other point process models, deep learning algorithms or traditional machine learning algorithms as the surrogate model for the distribution of the immediate feedback is not convincing. - Experiments were conducted by constructing two simulators but not the real human action records to perform the evaluation, making the experimental results not very convincing. - Source code of the proposed model is not publicly available, making the re-implementation of the proposed model challenging. - The proposed model is not time-aware, i.e., the proposed model doesn’t take time information into account. The most recent feedback may influence more on the model selection, compared to the out-of-date feedback.

Correctness: As far as I can see, there is nothing incorrect with the paper.

Clarity: This paper is generally well-written and structured well.

Relation to Prior Work: There are some previous works that integrating Gaussion process and learning (e.g., model selection). Some references on the integration of GP and learning algorithms can be included in the related work section.

Reproducibility: No

Additional Feedback: Code of the proposed model should be publicly available to the others.


Review 3

Summary and Contributions: This paper studies how to improve the efficiency of model selection for the production system. The authors propose an automated online experimentation mechanism (AOE) for model selection with few online experiments. They construct two synthetic experiments based on real data and demonstrate the effectiveness of the mechanism.

Strengths: 1. The authors study an important problem and the proposed model can efficiently perform model selection from a large pool of models with a small number of online experiments which is critical for the production system. 2. The authors propose using a Gaussian process to model the feedback, including a feedback (reward) model and a noise model which is a reasonable idea. 3. Authors model the uncertainty of the feedbacks which is also important to the production system.

Weaknesses: 1. The proposed Bayesian surrogate model contains two parts: 1) a GP model that captures the ``noise-free'' component of the immediate feedback, and 2) a noise distribution used to absorb all the stochasticity. It's not clear what are the noises? How do they affect the feedback? It would help better understand the model if authors can provide some examples. 2. For the uncertainty, the authors should explain what kind of uncertainty could be extracted from the noises. 3. The synthetic experiments look good, but it's not convincing if there is no real world experiments in production systems.

Correctness: Yes.

Clarity: The paper writing can be improved.

Relation to Prior Work: Yes.

Reproducibility: Yes

Additional Feedback:


Review 4

Summary and Contributions: The paper proposes a model selection algorithm called Model Selection with Automated Online Experiments (AOE) that is designed for use in production systems. In the problem statement, it is stated that the goal of the model selection problem is to select the model from a set of candidate models that maximises a metric of interest. It is assumed that the metric of interest can be expressed as the average immediate feedback from each of a model's predictions. AOE uses both historical log data and data collected from a small budget of online experiments to inform the choice of model. A distribution for the accumulative metric, or expected immediate feedback, is derived. It contains the distribution of inputs to the model, the distribution of model predictions conditioned on inputs and the distribution of the immediate feedback conditioned on inputs and predictions. The distribution of the immediate feedback is learned by a Bayesian surrogate model. The surrogate model is first trained on historical log data. Models are then selected sequentially for online experiments using an acquisition function. The data collected from each online experiment is used to update the surrogate model. This method is similar to Bayesian optimisation, but it is subtly different. Whereas Bayesian optimisation would use a surrogate model to model the metric conditioned on the choice of model, the surrogate model of the proposed method models the immediate feedback conditioned on inputs to the model and predictions returned by the model. There are model selection experiments in which AOE outperformed five baseline methods including Bayesian optimisation, which performed poorly in this setting. Contributions of the paper include: a derivation of the distribution of the accumulative metric; a method for approximating this distribution with a Bayesian surrogate model; a method for model selection for production systems that can utilise historical log data and data from online experiments.

Strengths: The idea of using both historical log data and online experimental data to inform the model selection process is novel. The method for approximating the distribution of the metric based on the surrogate model of the immediate feedback is also novel. There has been interest in model selection and related problems such as hyperparameter optimisation and neural architecture search, so it is likely that this paper would be of interest to the NeurIPS community. The experimental results are strong. In both the classification and recommender system experiments, AOE found models with an accumulative metric score that was closer to the optimal accumulative metric score. In the same experiments, AOE was able to predict the value of the accumulative metric with lower root mean squared error. The AOE method appears to be applicable to a wide range of model selection problems since it can be used with both binary immediate feedback and real-valued immediate feedback. The only restrictions are that the metric of interest must be expressible as the average of this immediate feedback. However, the paper explains that this restriction is not very restrictive and cites model selection for recommender systems as a use case where the click-through rate of users is the average of the immediate feedback.

Weaknesses: In the description of the AOE method it is stated that the immediate feedback can be real-valued or binary. However, there were no experiments where the immediate feedback was real-valued. It would have been nice to see an experiment with real-valued immediate feedback to verify that the AOE method still compares as favourably to the baseline methods when the immediate feedback is real-valued. The paper uses a Gaussian process (GP) as the surrogate model. Since the amount of historical log data is potentially large, the GP is a sparse GP where the predicted variance is computed with the FITC approximation. Based on the experimental results, the sparse GP appears to work well. However, a class of probabilistic model that scales to large amounts of data without such sparsification would perhaps be a more suitable choice for the surrogate model. For example, a paper from the NeurIPS 2016 conference called “Bayesian Optimisation with Robust Bayesian Neural Networks” demonstrated that Bayesian neural networks can be used as a surrogate model instead of GPs to allow Bayesian optimisation to scale to more function evaluations. Perhaps, a Bayesian neural network surrogate model would be appropriate in this setting as well. --- POST REBUTTAL --- I read the author response and the other reviews. The authors have successfully addressed my concerns about a few minor issues. I keep my initial view on the paper that this is an interesting, novel, and solid piece of work, as well as my initial score.

Correctness: After reading through the paper twice, the claims, the methods and the empirical evaluation all appear to be correct.

Clarity: The paper is very well written. The order of all the sections and subsections worked well. Each one followed on nicely from the last. The descriptions of the problem and the method were all clear.

Relation to Prior Work: The relation to prior work is clearly discussed. The paper discusses similarities and differences between AOE and two other approaches to model selection. The first is A/B testing, which like AOE uses online experiments to inform the choice of model, but unlike AOE does not use historical log data. The second approach is off-policy evaluation (OPE). OPE uses only historical log data to estimate the metric of interest for candidate models. The paper highlights issues with these two competing approaches that are addressed by AOE. Finding the best model with A/B testing may require many online experiments. However in practice, online testing of candidate models can be time-consuming or subject to resource constraints which means the number of online experiments that can be afforded is usually very small. On the other hand, OPE can predict the metric of interest very well for candidate models that behave similarly to the model used to collect the log data, but may struggle to do so when a candidate model behaves very differently to the log data model. To address these issues AOE combines the efficiency of OPE by first learning from historical log data and the more direct measure of a candidate model used in A/B testing by utilising a small number of online experiments. In addition, the paper mentions problems in the literature that share some similarities with the model selection problem. Among these are hyperparameter optimisation and neural architecture search.

Reproducibility: Yes

Additional Feedback:

[Author Response · NeurIPS 2020]

We thank the reviewers for the in-depth reviews. In this paper, we propose a new paradigm for model selection for production systems (MSPS), in which model selection is achieved by sequentially deploying a list of candidate models in order to discover the best model with the minimum number of online experiments. We show that a Gaussian process (GP)-based surrogate model can efficiently guide exploration-exploitation trade-off and outperforms a set of baseline methods. We will first answer the comments shared by multiple reviewers and then answer the individual comments.

**[R2 & R4] Non-GP surrogate.** As suggested by the reviewers, the surrogate models that are beyond Gaussian process (GP) have been studied in many related areas such as uncertainty quantification, Bayesian optimization, which often have non-stationality and better scalability. However, as we propose a new MSPS paradigm, the main focus of our work is to demonstrate: (i) sequential online deployment can lead to more efficient model selection than the state-of-the-art approaches (A/B tests and OPE). (ii) model uncertainty via acquisition functions can efficiently guide exploration-exploitation in our setting. Using a GP as the surrogate model allows us (i) to have a closed-form distribution of accumulative metric; (ii) to easily get high-quality uncertainty via being Bayesian non-parametric. Such a clean solution is important for developing and understanding a new method. Exploring other surrogate models that overcome the limitations of GP is beyond the scope of this paper and is an excellent direction for future research.

**[R2 & R3] Experiments on production system.** As reviewers 2 & 3 suggest, experiments on production systems can indeed reveal the true performance of our methods. However, we argue that such experiments are often noisy, conducted in an uncontrolled environment making a fair comparison against baselines practically impossible. On the contrary, the simulation studied in our paper provides valuable insights that are reproducible and helps gain a deep understanding of all methods presented. Such simulation studies are common in the recommender system and information retrieval research (Rohde et al. 2018; Chaney et al. 2018; Joachims et al. 2018; Jagerman et al. 2019). Furthermore, the results on production system will not be reproducible as public release of such datasets would not be possible due to IP / privacy concerns.

**[R1] Missing technical details.** We will add additional details about sparse GP, VI and binary observation in suppl.

**[R1] How is the probability matrix P computed?** A matrix $\mathbf{P}$ is computed for each model. Each column is associated with an input $\mathbf{x}_t$ and its values represents the probabilities of taking the corresponding actions (there are $K$ rows, one for each action, summed to one). We collect the inputs in all the online experiments so far and use them to compute $\mathbf{P}$.

**[R2] The proposed model is not time-aware...** As our method focus on allowing model selection from a large candidate pool with a limited A/B test bandwidth, we target at the scenarios where candidate models are deployed to A/B tests for a few weeks and then select the best model, which is the common scenario in industry. The state-of-the-art approaches such as A/B tests and OPE share the same challenges as our method regarding time sensitive systems. Model selection for a time sensitive system is an interesting and open research question for future work.

**[R2] Open sourcing.** We intend to release the source code if our submission is accepted.

**[R3] What are the noises in the noise distribution? How do they affect the feedback? ... what kind of uncertainty could be extracted from the noises.** For example, for a recommender system, the noise can be due to human's random behavior (randomly deciding to click a link) or the information that is not available to the system such as the mood of the moment or influence of another person. As these data variance cannot be explained by the input to the system, these data variance will be learned into the noise distribution. (For a Gaussian distribution, this will lead to a higher variance). This is often referred to as aleatoric uncertainty. This noise variance will be excluded when estimating the distribution of the accumulative metric (see Sec. 3.2), because these variance cannot be reduced by collecting more data.

**[R3] In line 113, ... what motivates a low-dimensional representation and/or why ... is good enough? Is the low dimensional representation still deterministic? Does it involve uncertainty?** Learning latent representations is a common technique for latent variable models such as variational auto-encoder. Being low dimension (5D in our case) allows the model to better determine the representations with a limited amount of user-item interactions. Thanks to the high modeling capability of GP (a universal approximator), even complex data like natural images can be encoded into low dimensional representations. In our model, the latent representations are in the form of variational posterior (not deterministic). The uncertainty in these variational posteriors contributes to the uncertainty of the accumulative metric, which is used to guide exploration-exploitation trade-off.

**[R4] Experiments with real-valued immediate feedback.** We choose to conduct experiments on binary immediate feedback, because it is widely used. Binary feedback is often easy to interpret and less biased (e.g., on a streaming platform using the time that a user spends can lead to a bias towards the content with longer duration.) We do not expect our method perform differently on real valued immediate feedback. We quickly run a small experiment (20 repeats) comparing our method with IS-g on a synthetic task with 10,000 candidate models (see the figure).



[Meta-Review · NeurIPS 2020]

This paper proposes an extension to Bayesian optimization methods for model selection. A surrogate model for the dataset is added to the setup, which allows the optimization to take more information to account as data is collected over time. The reviewers generally thought this was an interesting approach and an important direction. The main debate focused on the significance of the synthetic results based on real data, and whether they can be expected to generalize. We think that the clear novelty and the positive results outweigh this weakness.